# Human Archaeological Dentin as Source of Polar and Less Polar Metabolites for Untargeted Metabolomic Research: The Case of *Yersinia pestis*

**DOI:** 10.3390/metabo13050588

**Published:** 2023-04-24

**Authors:** Diego Armando Badillo-Sanchez, Donald J. L. Jones, Meriam Guellil, Sarah A. Inskip, Christiana L. Scheib

**Affiliations:** 1School of Archaeology and Ancient History, University of Leicester, Leicester LE1 7RH, UK; 2Leicester Cancer Research Centre, RKCSB, University of Leicester, Leicester LE1 7RH, UK; 3The Leicester van Geest MultiOmics Facility, University of Leicester, Leicester LE1 7RH, UK; 4Institute of Genomics, University of Tartu, 51010 Tartu, Estonia; 5Department of Evolutionary Anthropology, University of Vienna, 1030 Vienna, Austria; 6McDonald Institute for Archaeological Research, University of Cambridge, Cambridge CB2 3ER, UK; 7St. John’s College, University of Cambridge, Cambridge CB2 1TP, UK

**Keywords:** untargeted metabolomics, LC-HRMS, human dentin, biomolecular archaeology, disease, plague, ancient metabolomics

## Abstract

Metabolomic approaches, such as in clinical applications of living individuals, have shown potential use for solving questions regarding the past when applied to archaeological material. Here, we study for the first time the potential of this Omic approach as applied to metabolites extracted from archaeological human dentin. Dentin obtained from micro sampling the dental pulp of teeth of victims and non-victims of *Yersinia pestis* (plague) from a 6th century Cambridgeshire site are used to evaluate the potential use of such unique material for untargeted metabolomic studies on disease state through liquid chromatography hyphenated to high-resolution mass spectrometry (LC-HRMS). Results show that small molecules of both likely endogenous and exogenous sources are preserved for a range of polar and less polar/apolar metabolites in archaeological dentin; however, untargeted metabolomic profiles show no clear differentiation between healthy and infected individuals in the small sample analysed (*n* = 20). This study discusses the potential of dentin as a source of small molecules for metabolomic assays and highlights: (1) the need for follow up research to optimise sampling protocols, (2) the requirements of studies with larger sample numbers and (3) the necessity of more databases to amplify the positive results achievable with this Omic technique in the archaeological sciences.

## 1. Introduction

Organic material survives in and on a variety of archaeological materials and is a proven source of ancient biomolecules [1,2,3], the analysis of which has revolutionised our understanding of past peoples and societies. There has been significant work targeting macromolecules such as DNA and proteins in archaeological teeth, bones and dental calculus to assess factors such as past dietary intake [4,5], infectious diseases (reviewed in [6]), genetic ancestry (reviewed in [7]) and human bio-social structures (reviewed in [8]). DNA and proteomic studies of other archaeological and historical materials include manuscript parchments [9], ceramic pots [10,11], fossil egg shells [12] and clothing in museum collections [13].

Metabolites are small molecules (<1.5 kilo Daltons (kDa)) produced as by-products of cellular metabolism and can be studied using several approaches including Fourier-transform infrared (FTIR), mass spectrometry (MS) or nuclear magnetic resonance (NMR) spectroscopy, among several others. They can be found in any cellular material or material with cellular contact including blood, urine, hair, bone and soft tissue (for example see [14,15,16]). Metabolomic studies can be either targeted or untargeted and are a primary focus for disease biomarker discovery (reviewed in [17]). They have also proven useful for detecting elements of dietary intake in living individuals [18,19,20] as well as one study in pig bones [21].

Given the potential of metabolomics, archaeological scientists have been keen to assess the potential of the approach to expand our understanding of infectious and non-infectious diseases, medicinal practice and diet in the past, as they offer an opportunity to obtain higher resolution data and to target multiple types of molecules at once. Metabolomics-based approaches have been used to address such questions as the use of intoxicants as evidenced by residues in smoking pipes [22,23] and in human hair [24,25,26], to characterise the gut microbiome composition in mummies [27], to reunite a mummy with its sarcophagi [28] and to identify ancient grains [29]. To date, however, there has been extremely limited research on human skeletal remains (HSR) with only two studies using mineralised dental plaque (calculus) [30,31] and one on post-cranial osteological material [32]. This latter study has been particularly important in demonstrating that thousands of molecules can be detected in HSR and that there are potentially differences in the metabolome between individuals of different disease states, even though bone is susceptible to diagenesis and change while in the ground. 

The metabolome of modern dental pulp has been studied in living individuals in both healthy and diseased states [33] and that of dentin has been investigated for characterising the foetal and early childhood exposome [34]. Archaeological dentin thus remains a novel potential source of ancient metabolomic information. Teeth, in general, preserve DNA better than less dense post-cranial materials [35] such as those used in [32] and the dental pulp chamber is a highly vascularised, proven source of endogenous human DNA and proteins and exogenous, infectious pathogens [36]. As a record of the individual’s bloodstream, ancient dental pulp could reveal intake of dietary sources, medicinal compounds and/or intoxicants as well as non-communicable and infectious disease biomarkers. 

Plague is an infectious disease caused by a Gram-negative bacterium, *Yersinia pestis*, and has swept through human populations in different pandemics for at least 5000 years [37] and can kill as rapidly as within 24 h or several days depending on the type (pneumonic vs. bubonic), infection source (respiratory vs. flea bite) and individual health conditions [38,39]. It has been referred to as an indiscriminate killer; however, different studies of historical pandemics have shown differential estimated mortality rates for different social subgroups [40,41]. Understanding whether inherited (genomic) or cultural (diet) differences led to a susceptibility to plague would increase our understanding of the impact of this pathogen on human populations. 

Using an ancient population that lived through the Plague of Justinian (541–549 CE), this study investigates whether (1) metabolites can be retrieved from ancient human dental pulp drilled from inside of archaeological human teeth, (2) any difference exists between victims (detected *Y. pestis* ancient DNA (aDNA) in teeth) and putative non-victims (no detected *Y. pestis* aDNA in teeth) and (3) any detected metabolite(s) or metabolome differences could be used as markers of dietary or medicinal intake, or biomarkers for *Y. pestis* infection. 

## 2. Materials and Methods

### 2.1. Archaeological Site

The site of Barrington A, Edix Hill, sits approximately 5 km southwest of Cambridge, was in use from at least 500 CE to 650 CE [42] and has been well studied in recent years [43,44,45]. The unexpected discovery of *Y. pestis* at the site [46] opened new pathways for investigation into the spread and impact of the Plague of Justinian (541–549 CE), part of the First Pandemic (541–750 CE). Using metagenomic screening of aDNA from tooth roots, at least ten victims of plague were identified (publication in preparation). 

Twenty individuals were selected for the untargeted metabolomic study (Table 1) according to the following: first, 100 individuals from the site of Edix Hill were screened for human and pathogen DNA (publication in prep) using an established pipeline (see [43]); of these, ten individuals with a positive identification of ancient *Yersinia pestis* DNA in a tooth were chosen, then each individual was matched by estimated morphological age and genetic sex to an additional individual that did not have any ancient *Y. pestis* DNA identified.

### 2.2. Dentin Sampling

The same teeth used for the metagenomic screening were micro-sampled by drilling the open bottom of the root canal (of the root used for the screening) using a sterilised engraving drill tip and stored in a DNA lo-bind 1.5 mL tube. Masses varying between 14 and 65 mg were obtained depending on the tooth. Samples with enough mass were split into three to obtain biological replicates for the metabolomic study for a total of 42 samples as indicated in Table 1. 

### 2.3. Extraction of Polar and Less Polar/Apolar Metabolites from Dentin Samples

For the metabolite extraction procedure on the archaeological dentin powder, a liquid–solid process was performed as follows: (1) 13 mg for each sample (including replicates) was placed in a pre-labelled 1.5 mL plastic tube with an O-ring cap which holds four ceramic beads of 3 mm of diameter (zirconium oxide). (2) A total of 100 µL of cold (4 °C) methanol was added, the tubes sealed and placed in a bead blaster tissue homogeniser for a sequence of 12 repetitions at 600 Hz and cycles of 25 s of agitation and 25 s of pause. (3) Tubes were centrifuged at 16,000 RCF for 5 min at 10 °C and then supernatant transferred to a new pre-labelled 0.5 mL plastic tube. (4) Steps 2 and 3 were repeated on the same initial tube by adding each time 100 µL of ethanol, 100 µL of 0.1 % formic acid and 100 µL of a solution 75:25 water:methanol, combining all the supernatants in the same tube. (5) A total of 20 μL of cold (4 °C) methanol was added to the extract solution and centrifuged at 20,000 RCF for 10 min at 10 °C. (6) The final supernatant was split into two new tubes. One tube was labelled as “A” for less polar/apolar studies and the second as “B” for polar studies. All extracts (A and B) were dried by first drying them for 2 h in a speed vacuum and then once overnight through a freeze dryer. 

### 2.4. Untargeted Metabolic by High-Flow-UPLC-IM-TOF-HRMS

For the different untargeted metabolomic assays, the different extracts were reconstituted as described elsewhere [32]. Untargeted analyses were performed on a high-flow Acquity UPLC system coupled to a Waters Synapt G2 HDMS system (Waters Corporation, Manchester) composed of an electrospray ionisation (ESI) source (operated either in positive mode or negative mode), an ion mobility (IM) cell and a time-of-flight detector (TOF). MS data were collected using the mobility MSe function in profile mode (full scan) over the *m*/*z* 50–1500 range. Capillary voltage was set at 3.00 kV, sampling cone 30 V, source temperature 120 °C, desolvation temperature 600 °C and desolvation gas 1000 L/h. IM and TOF cells were calibrated in advance and lock spray of leucine enkephalin was used and infused at 10 μL/min. Chromatographic separations were carried out by using both a reversed phase (RP) C18 column (Waters Acquity UPLC BEH C18 column (2.1 × 100 mm, 1.7 μm)) for semi polar and apolar metabolites, and a hydrophilic interaction liquid chromatography column (Acquity UPLC BEH HILIC (2.1 × 100 mm, 1.7 μm)) for polar metabolites. Samples were maintained at 4 °C in the autosampler and aliquots of 5 μL injected per run. The flow rate was set at 0.4 mL/min, and the column temperature was maintained at 40 ± 2 °C. Solvents used for separation consisted of 0.1% formic acid in water as mobile phase A, and 0.1% formic acid in acetonitrile as mobile phase B. The RP gradient elution program started running at 2% B, held for one minute, then mobile phase B was linearly increased to 20% to minute 5, 25% in minute 6, 75% in minute 7 and 98% in minute 7.5, then held at 98% of B until minute 8, going back to initial conditions of 98% of A in minute 9 and the column was equilibrated until minute 10. The HILIC gradient elution program started running at 3% A and held for one minute, then mobile phase A was linearly increased to 20% in 3 min, 25% in one minute, 75% in two minutes and 97% in one minute, then was held for half a minute, going back to initial conditions of 97% of B in 1.5 min to later be equilibrated for four minutes.

### 2.5. Metabolomic Analysis 

Quality assurance, data treatment, statistical analysis and metabolite putative annotation for metabolomic analysis were performed as described elsewhere [32].

#### 2.5.1. Quality Assurance (QA)

QA procedures were performed to reduce the unwanted variation regarding the pre-analytical, analytical and post-analytical phase during all the metabolomics workflow [47]. Blank injections of pure solvent were run to condition the LC-MS system and obtain the MS data for later data processing. Extraction blank injections were performed to evaluate the extraction process and determine the MS data which could interfere with the biological information from the different samples. Pooled quality control (QC) samples were prepared by mixing 5 µL of each extracted sampled after their reconstitution. QC samples were injected at the beginning of the sequence to stabilise the MS instrument, as well as bracketing the biological samples during the total sequence after each 6–8 randomised samples to monitor the stability of the analytical platform. To avoid instrumental and statistical bias, all biological samples were randomised before injection. A quality control standard mixture (LC-MS QC STD Part number 186006963-1, Waters) was used to test the technical variability and possible carry over of the instrument during the injection of the different sequences, being the r.t. and ions evaluated at the beginning and at the end of each sequence.

#### 2.5.2. Data Treatment

Progenesis-QI (Nonlinear dynamics) software was employed for preprocessing of each workflow (C18 or HILIC assays), performing the run alignment, ion detection, peak picking, isotope and adduct deconvolution, ion drift measurement, etc., on the raw data of all biological and QA samples, i.e., pooled QC, extraction blanks and injection blanks. Resultant compound metabolomic matrices containing the different samples and the total list of compounds detected for each experiment were inspected manually by using Microsoft Excel 2016 to filter out the non-biological information (noise, instrumental signals, compounds from the extraction process, etc.) For that, compounds with a contribution higher than 5% of the blank or extraction blank with respect to the QCs were filtered out. Furthermore, the compounds with a relative standard deviation (RSD) for the QC samples greater than 40% were removed. Before any further statistical analysis, data transformation was applied to the different data matrices. K-nearest neighbours based on similar features was employed to replace missing values and mean sample normalization; log10 data transformation; and Pareto data scaling were applied to the different sets of data.

#### 2.5.3. Statistical Analysis

MetaboAnalyst 5.0 (https://www.metaboanalyst.ca/, accessed on 2 August 2022) was employed for the statistical analysis of the untargeted metabolomic assays [48]. Multivariate statistical analyses, including unsupervised principal component analysis (PCA), were used to establish the overall differences in the metabolic profiles between groups studied. The quality of the established statistical models was evaluated using the R2X, R2Y and Q2 parameters, as well as permutation tests to evaluate the possible overfitting in supervised models. Statistical models were performed including QC samples to evaluate stability of the analytical system and the data quality, using as indicator the clustering of QC samples in successive injections during the sequences. Univariate tests were performed to visualise significant features up-regulated or down-regulated and volcano plots were drawn by transferring the fold change (FC) value of each substance peak to log2 (FC) and transferring the *p* value (*p* = 0.05) of Student’s *t*-test to −log10 (*p* value). Hierarchical clustering analysis was performed through a Euclidian distance measure and a Ward clustering algorithm, and the result presented as a hierarchical clustering dendrogram.

#### 2.5.4. Metabolite Putative Annotation

Putative annotation of the compounds found by the high-flow UPLC IM-TOF-HRMS measurements on C18 experiments was performed using the data identification window of the Progenesis QI software. Data were compared considering all potential matches using the following libraries: (1) METLIN MS/MS library 2017 for Progenesis QI plugin with a 3-ppm precursor and fragment tolerance for H = 0–150; C = 0–100; N = 0–10; O = 0–30; F = 0–4; Mg, P, S, Cl, Br, I = 0–2. (2) LipidBlast library plugin with a 3-ppm precursor tolerance and 10-ppm fragment tolerance. (3) Metabolic Profiling CCS Library search plugin with a 3-ppm precursor tolerance, 10 ppm fragment tolerance and 10% CCS tolerance.

## 3. Results

### 3.1. Less Polar/Apolar Untargeted Metabolomic Profile

After data pre-processing of the measurement followed through the LC-HRMS system by the C18 reversed phase column, 635 compounds were detected in the dentin extracts. By using the different QA samples and procedures, a total of 137 small molecules which can be attributable to less polar/apolar metabolites were found. It is important to highlight that it is not possible to determine at this stage if those correspond exclusively to the biological condition of the individuals in a perimortem state or because of taphonomic and diagenetic processes suffered by the burials in post-mortem conditions. All results are listed in Annex 1 in Appendix B. 

To evaluate the quality of the untargeted metabolomic assays for the less polar/apolar metabolites obtained, a PCA model was performed including the biological samples and the pooled QC samples after data filtering (Appendix A). The QC samples were located together in a similar region of the score plot; low dispersion suggests that the analytical performance of the assay was optimal, with slight differences attributable to the normal decay of the MS machine after a series of injections. No tendencies or anomalies were evident, suggesting that data can be treated for further metabolomic investigations. Next, to study the untargeted metabolomic profile of the different individuals a PCA model was made with the normalised data matrix of the different individuals (Figure 1). It was evident that samples do not cluster according to the classes expected (positive or negative *Y. pestis*), nor to the biological sex of the individuals. On the contrary, samples are dispersed and mixed, reflecting that the metabolomic profile in the different individuals is similar. Further detail in the results for single individuals and their biological replicates shows that replicates preserve the same tendency in the PCA score plot, but a variance is present. This can be explained as a consequence of the heterogeneous nature of the sample, where the content of total dentine can vary in the percentage of organic to inorganic composition, which suggests that further improvements to the sampling protocols are required to decrease this variance. 

A PCA score plot of the metabolomic profile for the less polar/apolar metabolites of the dentin extracts for the different individuals studied shows that samples tend to cluster in four different groups. A first group, located towards negative PC1 and positive PC2 values, is composed of three male child individuals (604, 607 and 623) of which two were positive for *Y. Pestis*. A second group is composed of five individuals (609, 652, 547, 618 and 590), three females and two males of variable young age, with positive and negative results to the *Y. pestis* condition. A third group can be associated with four individuals located around the centre of the PCA score plot (625, 587, 576 and 633), three females of a variable young age and a child male which has a tendency in the direction of the first group. A fourth group located towards positive PC1 and both positive and negative PC2 values is made of eight individuals (558, 575, 598, 554, 582, 603, 589 and 583) of variable young age, equal number of males to females and almost 63% of positive cases for *Y. pestis.* The dispersion among observations obtained in this PCA model does not allow us to conclude that the less polar/apolar metabolites present on the dentin samples of the infected and control individuals are different, suggesting that at this stage no differences due to the disease state can be found in the samples. 

### 3.2. Polar Untargeted Metabolomic Profile

Untargeted metabolomic assay of the polar compounds present in the dentin extracts by LC-HRMS after HILIC column separation show a total of 2576 compounds detected after pre-processing the raw data. After data treatment and QA procedures, 493 polar compounds were obtained with possible biological significance. All results are listed in Annex 2. 

To evaluate the MS data, a PCA model was made including both QC samples and biological samples (Appendix A). The score plot of the PCA model showed no evident separation for the different class groups studied here and observations from the QCs had a low variance among them. No evidence of analytical problems or any type of singularity with the data was observable, indicating that the data can be used for further statistical analysis. 

After processing and data normalisation of the untargeted metabolomic matrix for the polar compounds, a PCA model was made with the biological samples (Figure 2A). The score plot of the PCA model shows a slight tendency to positive PC2 values for positive *Y. pestis* individuals, but no clear separation between the infected and healthy groups. Evaluation of the replicates by individuals shows that some samples have larger dispersion between replicates for a single individual, being dispersed around the score PCA plot without an overall tendency. To better evaluate the similarity between samples and replicates, a hierarchical clustering test was performed, and results evaluated according to the dendrogram presented in Figure 2B. There it was evident that some replicates (590_1, 623_1, 623_2, 623_3, 633_3, 618_1, 652_1, 652_2, 652_3, 625_2, 598_3, 576_1, 576_2, 576_3 and 607_1) presented variances that could affect the statistical result in the overall set of samples. For that reason, to better explore the data, a second PCA model was performed after removing those samples which presented larger variances (Figure 2C).

The second PCA model created after removing the samples with higher variance showed better separation between the two class groups studied, where those individuals in the infected group presented a tendency to have more positive PC2 values. From the score PCA plot five individuals are highlighted (590, 618, 582, 625 and 589) that cluster in a group with negative PC1 values and positive PC2 values. Univariate statistical analysis of the different metabolites shows a total of 26 metabolites presented statistical differences among the groups, 11 significantly higher in the positive group and 15 significantly lower in the positive group with respect to the negative group. Results are presented in Figure 3 and Appendix A.

### 3.3. Putative Annotation

The different compounds obtained after the untargeted metabolomic LC-HRMS assays were subjected to putative annotation processes (Annex 1 and 2). Comparison for both polar and less polar/apolar metabolites with the different spectral libraries presented a low number of molecules able to be annotated. The major part of them remain as unknowns (Appendix A).

## 4. Discussion and Conclusions

The aims of this study were to establish whether: (1) metabolites can be extracted from ancient dentin, (2) any statistically significant differences exist between infected and non-infected individuals and (3) any metabolite(s) could be used as a biomarker of plague infection. Metabolites can be retrieved from ancient human dental pulp chambers. Polar compounds appear to be more persistent and in higher numbers than apolar/less polar compounds. The less/apolar metabolome results showing similar profiles across all samples could be due either to a true signal of homogeneity in the sample source or, more likely, due to endogenous compounds being more likely to be degraded and removed during the normal taphonomic process and replaced with new compounds from the environment. Incursion of the dental material by exogenous sources originating in the soil is pervasive (Annex 1 and 2). Compounds that were identified (Appendix A) mainly originate in bacteria or plants that live in soil. We are unable to distinguish between endogenous and exogenous origin in this study because no soil samples from the burials were analysed due to the excavations being carried out in the late 1980s/early 1990s. The individuals sampled in this study were buried across the excavation site with no apparent clustering of plague victims and putative non-victims [42]. Thus, the signal is unlikely to be due solely to environmental conditions. 

For the polar metabolites, though there seems to be a slight tendency for separation between infected and non-infected individuals, it is not currently possible to distinguish between victims and non-victims by PCA. Several compounds were identified as statistically significantly up- or down-regulated between the groups; however, further research is needed to identify those compounds (Appendix A) and whether these are related to pathogen metabolism, host immune reaction or another host-mediated factor. 

In terms of a metabolite biomarker, in the specific case of the disease here investigated, a sample size and power analysis test was performed and results are presented in Appendix A. In order to obtain reliable metabolites that could be used as biomarkers for the detection of *Y. pestis* infection in dentin material, a study with more than 700 individuals will be required to obtain a 95% probability and 70% prediction power. Thus, the current method of using metagenomics to screen for *Y. pestis* aDNA remains the easier and more efficient method for archaeological research. Though it is important to note that DNA is not perfect, the problem of false negatives due to preservation conditions remains important. For research questions where available material does not allow for high sample numbers, a targeted approach is more likely to be successful than a shotgun approach. One study of plasma metabolites in non-human primates exposed to pneumonic plague revealed changes in metabolites related to lipid pathways, indicative of inflammation and oxidative stress [49]; however, a ‘plague metabolome’ has not been characterised, precluding a targeted study at this time. The results of this study show promise for further investigations, particularly into optimisation of extraction methods, for metabolomics studies in archaeological dentin by LC-HRMS measurements of polar metabolites. 

## Figures and Tables

**Figure 1 metabolites-13-00588-f001:**
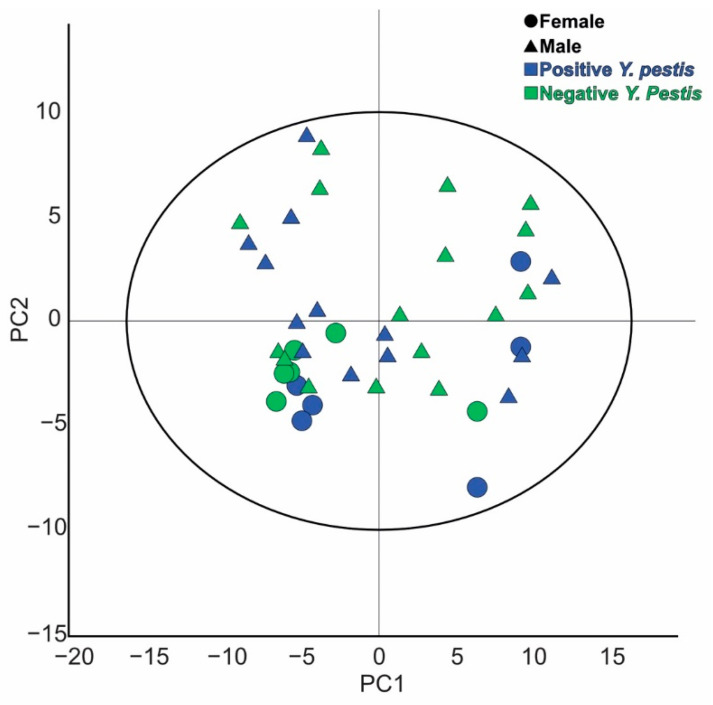
PCA scatter plot for the normalised data obtained after LC-HRMS measurements in the untargeted metabolomics assay for less polar/apolar metabolites through reversed phase C18 column separation. Model made by 41 observations, 137 variables. PC1 R^2^X: 0.291, Q^2^: 0.24; PC2 R^2^X: 0.11, Q^2^: 0.0646. Biological sexes are represented as triangles for males and circles for females. Negative *Y. pestis* individuals are coloured in green and positive individuals in blue.

**Figure 2 metabolites-13-00588-f002:**
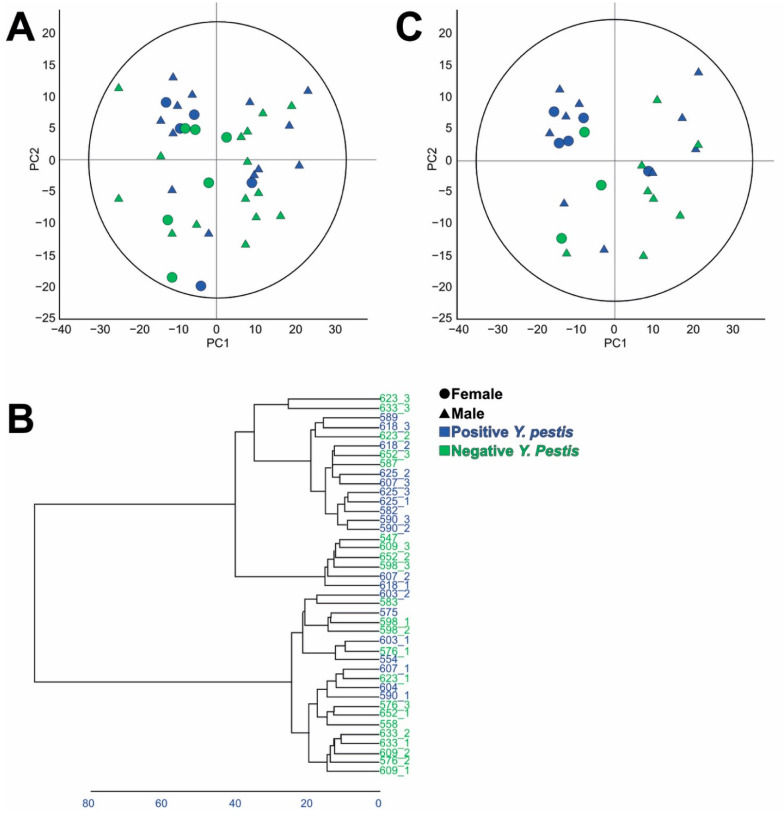
(**A**) PCA scatter plot for the normalised data obtained after LC-HRMS measurements in the untargeted metabolomics assay for polar metabolites through HILIC column separation with all observations included. Model made by 41 observations, 493 variables. PC1 R^2^X: 0.328, Q^2^: 0.282; PC2 R^2^X: 0.146, Q^2^: 0.177. (**B**) Dendrogram for the different samples in the untargeted metabolomics assay for polar compounds (**C**) PCA scatter plot for the normalised data obtained after LC-HRMS measurements in the untargeted metabolomics assay for polar metabolites through HILIC column separation with selected samples. Model made by 26 observations, 492 variables. PC1 R^2^X: 0.354, Q^2^: 0.287; PC2 R^2^X: 0.142, Q^2^: 0.151. Biological sexes are represented as triangles for males and circles for females. Negative *Y. pestis* individuals are coloured in green and positive individuals in blue.

**Figure 3 metabolites-13-00588-f003:**
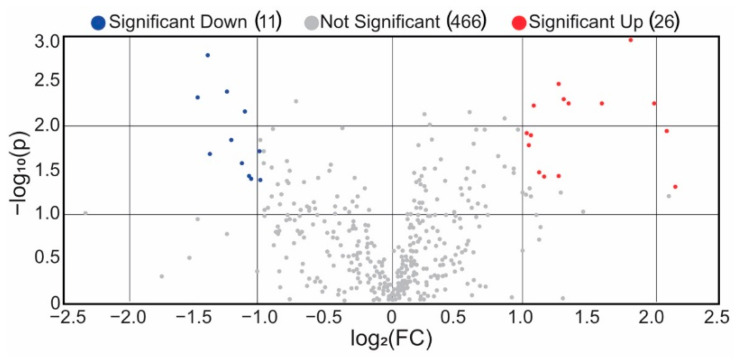
Volcano plot for the polar metabolites detected through the untargeted metabolomics assay after removal of high variance replicates; each dot represents one metabolite. Blue dots are significantly down-regulated metabolites. Red dots are significantly up-regulated metabolites. Grey dots are non-significant metabolites.

**Table 1 metabolites-13-00588-t001:** List of archaeological samples used for the untargeted metabolomic study of human dentin. Genetic sex is indicated as XX for female and XY for male.

Lab ID	Grave	Sk ID	Sample	Age from Site Report	*Y. pestis* DNA	Genetic Sex	Extraction ID
547	55	171	LRM3	Adult	Negative	XX	26
554	76	405	URI1	Young Adult	POSITIVE	XY	7
558	84	440A	LRI2	Young Adult	Negative	XX	35
575	78	424	LRI2	Juvenile	POSITIVE	XX	15
576	77	423	URPM1	Young Adult	Negative	XY	3, 18, 21
582	96	547B	ULC	Late child	POSITIVE	XX	36
583	97	551	LRI1	Young Adult	Negative	XY	19
587	102	586	LLI2	Child	Negative	XX	24
589	106	626A	LRI2	Young Adult	POSITIVE	XX	40
590	106	626B	URI2	Young Adult	POSITIVE	XY	8, 34, 38
598	2	3C	URI1	Young Adult	Negative	XY	13, 16, 23
603	73	372	LLM3	Young Adult	POSITIVE	XY	6, 10, 33
604	85	447B	dULC	Child	POSITIVE	XY	5
607	94	529	LRM1	Child	POSITIVE	XY	2, 17, 22
609	112	719A	LRM3	Adult	Negative	XY	4, 11, 25
618	114	726	URM3	Young Adult	POSITIVE	XX	20, 37, 39
623	18	42A2	dU?LC	Child	Negative	XY	1, 32, 42
625	85	447A	ULI1	Young Adult	POSITIVE	XY	29, 30, 31
633	36	125	LLC	Young Adult	Negative	XY	9, 12, 41
652	10	16B	ULI2	Young Adult	Negative	XX	14, 27, 28

## Data Availability

The different RAW data generated during the current study are available in the Metabolights repository under the reference MTBLS7394 (www.ebi.ac.uk/metabolights/MTBLS7394, accessed on 2 March 2023) [50].

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
