# Peer review of "Human Archaeological Dentin as Source of Polar and Less Polar Metabolites for Untargeted Metabolomic Research: The Case of Yersinia pestis"

_metabolites, 2023, doi:10.3390/metabo13050588_

Round 1

Reviewer 1 Report

In this manuscript, Badillo-Sanchez et al. investigated if metabolites can be extracted from ancient dentin and be used as a biomarker of plague infection. Authors successfully retrieved metabolites from ancient human dental pulp chambers and the polar metabolites might show a slight tendency between infected and non-infected individuals. Overall, the manuscript was written well and ready to be published. 

Author Response

Reviewer #1: 

In this manuscript, Badillo-Sanchez et al. investigated if metabolites can be extracted from ancient dentin and be used as a biomarker of plague infection. Authors successfully retrieved metabolites from ancient human dental pulp chambers and the polar metabolites might show a slight tendency between infected and non-infected individuals. Overall, the manuscript was written well and ready to be published. 

Thank you for your comments.

Reviewer 2 Report

In the manuscript ID: metabolites-2315186, the authors use an untargeted metabolomic approach to assess the possibility of finding metabolites in the ancient dentin and of using them as putative markers to infer pathogenic death causes, i.e., plague. The obtained results, despite preliminary and needing future improvements, indicate the use of polar metabolites as possible markers to be adopted, together with the analysis of the ancient DNA, in the archaeological research.

 The study is overall interesting and sounding in the extent of a brief communication. The introduction provides a comprehensive background of the topic, allowing the reader to easily understand the aims of the study. The methods are clear and detailed; even if reported in a cited reference, it is advisable to list the statistical analysis procedures adopted to analyse the obtained data. The results are overall sounding and well commented in the discussion.

A few minor concerns:

-In Table 1 same age ranges, attributed to different individuals (adults or young adults) are overlapping. Could the authors please clarify on it?

-Lines 208 and 244, please type Y. pestis in italic.

After performing these minor modifications, the paper can be considered for publication in “Metabolites”.

Author Response

Reviewer #2: 

In the manuscript ID: metabolites-2315186, the authors use an untargeted metabolomic approach to assess the possibility of finding metabolites in the ancient dentin and of using them as putative markers to infer pathogenic death causes, i.e., plague. The obtained results, despite preliminary and needing future improvements, indicate the use of polar metabolites as possible markers to be adopted, together with the analysis of the ancient DNA, in the archaeological research.

The study is overall interesting and sounding in the extent of a brief communication. The introduction provides a comprehensive background of the topic, allowing the reader to easily understand the aims of the study. The methods are clear and detailed; even if reported in a cited reference, it is advisable to list the statistical analysis procedures adopted to analyse the obtained data. The results are overall sounding and well commented in the discussion.

A few minor concerns:

1) In Table 1 same age ranges, attributed to different individuals (adults or young adults) are overlapping. Could the authors please clarify on it?

Age estimation in skeletal remains is difficult and many systems exist. These labels and ages were taken directly from the archaeological site report. We agree it is confusing and have updated it by removing the ages in brackets as these numbers are subjective to the observer and not necessarily accurate. We have kept just the general age categories. 

2) Lines 208 and 244, please type Y. pestis in italic.

Corrected. 

After performing these minor modifications, the paper can be considered for publication in “Metabolites”.

Reviewer 3 Report

The authors performed untargeted metabolomics analysis from ancient dentin samples. The samples are rare, and the analytical techniques are up to date.

1) Line 121 and 126, centrifugation rate should be in g not RPM unit.

2) section 2.5 needs to be added with detailed information on metabolomics data analysis (even though one paper has been cited here), especially data extraction, metabolite annotation, data normalization, and statistical analysis (e.g. how dendrogram and PCA was performed).

3) percentage of variance represented by each PC should be shown in the figures.

4) to make a more meaningful result, suggest to annotate the putative annotated compounds with significance (those highlighted in figure 3), by running a second LCMS with retention time markers or standards. And discussion could be based on those metabolites. The current manuscript is not exhibiting significant results from this analysis.

Author Response

Reviewer #3: 

The authors performed untargeted metabolomics analysis from ancient dentin samples. The samples are rare, and the analytical techniques are up to date.

1) Line 121 and 126, centrifugation rate should be in g not RPM unit.

We apologize for the mistake, and thanks for the correction, the units should be in RCF but were noted as RPM. Units were corrected in the document.

2) section 2.5 needs to be added with detailed information on metabolomics data analysis (even though one paper has been cited here), especially data extraction, metabolite annotation, data normalization, and statistical analysis (e.g. how dendrogram and PCA was performed).

We appreciate this suggestion, the original manuscript is intended as a short manuscript, for that reason we had referenced the detail of the metabolomic steps. But as suggested by the reviewer the complete metabolomics process has been included in detail in section 2.5.

3) percentage of variance represented by each PC should be shown in the figures.

The statistical information is included in the caption of the original figures as PC1 R2X and PC2 R2X.

4) to make a more meaningful result, suggest to annotate the putative annotated compounds with significance (those highlighted in figure 3), by running a second LCMS with retention time markers or standards. And discussion could be based on those metabolites. The current manuscript is not exhibiting significant results from this analysis.

We appreciate the reviewer points.

The putative annotation process if done well could be a plus for any metabolomic manuscript; however, we feel that the scope of such an extensive work won't have a deep meaning in the present study, due to the minor separation of the metabolomic profiles between the two groups here studied. The proper annotation of the molecules would not bring changes to the goal to obtain better descriptors of the condition of interest. This implies a long effort in time, MS and bioinformatic resources that will still have a low impact.

In the same way, as the archaeological dentin results showed a variability due to factors such as micro sampling and selection of the teeth to test on the untargeted approach applied, confidence in the noted compounds will be low, even if a high annotation level is achieved, presenting a risk for the possible archaeological interpretations.

We agree that statistical differentiation between the metabolomic profile of the two groups studied here is not achieved, nevertheless, results about the possibility to perform studies in such rare material (archaeological dentin) and the fact that is proven that ancient small molecules are preserved in this material, are relevant and significant results that will improve our fields of study after being discussed by the academic community.

Future experiments following the findings reported here are planned with the aim to continue the exploration of such unique samples and in the future full research manuscripts will be produced. At the present time, we consider this brief report to be vital for our work and other researchers in our field.

Round 2

Reviewer 3 Report

Thanks for the responses.